# Chiral singlet superconductivity in the weakly correlated metal LaPt$_3$P

P. K. Biswas [1,9 ✉], S. K. Ghosh [2,9 ✉], J. Z. Zhao[3], D. A. Mayoh [4], N. D. Zhigadlo [5,6], Xiaofeng Xu [7], C. Baines[8], A. D. Hillier[1], G. Balakrishnan [4] & M. R. Lees [4]

Chiral superconductors are novel topological materials with finite angular momentum Cooper pairs circulating around a unique chiral axis, thereby spontaneously breaking time-reversal symmetry. They are rather scarce and usually feature triplet pairing: a canonical example is the chiral $p$-wave state realized in the $A$-phase of superfluid He$^3$. Chiral triplet superconductors are, however, topologically fragile with the corresponding gapless boundary modes only weakly protected against symmetry-preserving perturbations in contrast to their singlet counterparts. Using muon spin relaxation measurements, here we report that the weakly correlated pnictide compound LaPt$_3$P has the two key features of a chiral superconductor: spontaneous magnetic fields inside the superconducting state indicating broken time-reversal symmetry and low temperature linear behaviour in the superfluid density indicating line nodes in the order parameter. Using symmetry analysis, first principles band structure calculation and mean-field theory, we unambiguously establish that the superconducting ground state of LaPt$_3$P is a chiral $d$-wave singlet.

[1] ISIS Pulsed Neutron and Muon Source, STFC Rutherford Appleton Laboratory, Harwell Campus, Didcot, Oxfordshire, UK. [2] School of Physical Sciences, University of Kent, Canterbury, UK. [3] Co-Innovation Center for New Energetic Materials, Southwest University of Science and Technology, Mianyang, China. [4] Physics Department, University of Warwick, Coventry, UK. [5] Laboratory for Solid State Physics, ETH Zurich, Zurich, Switzerland. [6] CrystMat Company, Zurich, Switzerland. [7] Department of Applied Physics, Zhejiang University of Technology, Hangzhou, China. [8] Laboratory for Muon Spin Spectroscopy, Paul Scherrer Institute, Villigen PSI, Switzerland. [9] These authors contributed equally: P. K. Biswas, S. K. Ghosh. ✉email: pabitra.biswas@stfc.ac.uk; S.Ghosh@kent.ac.uk

ooper pairs in conventional superconductors (SCs), such as the elemental metals, form due to pairing of electrons by phonon-mediated attractive interaction into the most symmetric s-wave spin-singlet state[1]. They have a nonzero onsite pairing amplitude in real-space. In contrast, unconventional SCs are defined as having zero onsite pairing amplitude in real-space[2]. As a result, electrons in Cooper pairs of unconventional SCs avoid contact with each other to become energetically more favourable over conventional Cooper pairs, in strongly repulsive systems. Unconventional SCs pose a pivotal challenge in resolving how superconductivity emerges from a complex normal state. They usually require a long-range interaction[3] and have lower symmetry Cooper pairs.

Chiral SCs belong to a special class of unconventional SCs having non-trivial topology and Cooper pairs with finite angular momentum. A well established realization of a chiral p-wave triplet superconducting state is in the A-phase of superfluid He3 [4]. In bulk materials, perhaps the best studied examples are UPt3[5] and Sr2RuO4[6]. The long-held view of Sr2RuO4 being a chiral p-wave triplet SC[7], however, has been called into question by recent NMR[8] and neutron[9] measurements, and a multicomponent chiral singlet order parameter has been suggested to be compatible with experiments[10]. UPt3 is believed to realize a chiral f-wave triplet state, although many open questions still remain[7]. Recently, the heavy fermion SC UTe2 has been proposed to be a chiral triplet SC[11]. Chiral singlet SCs are also extremely rare, but may be realized within the hidden order phase of the strongly correlated heavy fermion SC URu2Si2[12] and in the locally noncentrosymmetric material SrPtAs[13] although there are many unresolved issues for both these materials.

LaPt3P is a member of the platinum pnictide family of SCs APt3P (A = Ca, Sr and La) with a centrosymmetric primitive tetragonal structure[14]. Its $T_c = 1.1$ K is significantly lower than its other two isostructural counterparts SrPt3P ($T_c = 8.4$ K) and CaPt3P ($T_c = 6.6$ K)[14], which are conventional Bardeen-Cooper-Schrieffer (BCS) SCs. Indications of the unconventional nature of the superconductivity in LaPt3P come from a number of experimental observations: a very low $T_c$, unsaturated resistivity up to room temperature and a weak specific heat jump at $T_c$[14]. LaPt3P also has a different electronic structure from the other two members in the family because La contributes one extra valence electron. Theoretical analysis based on first principles Migdal-Eliashberg-theory[15,16] found that the electron–phonon coupling in LaPt3P is the weakest in the family, which can explain its low $T_c$. The weak jump in the specific heat which is masked by a possible hyperfine contribution at low temperatures[14] (see also Supplementary Fig. 2), however, cannot be quantitatively captured.

Here, we show that the weakly correlated metal LaPt3P spontaneously breaks time-reversal symmetry (TRS) in the superconducting state at $T_c$ with line nodal behaviour at low temperatures based on extensive muon-spin relaxation ($\mu$SR) measurements. Using first principles theory, symmetry analysis and topological arguments, we establish that our experimental observations for LaPt3P can be consistently explained by a chiral d-wave singlet superconducting ground state with topologically protected Majorana Fermi-arcs and a Majorana flat band.

## Results

We have performed a comprehensive analysis of the superconducting properties of LaPt3P using the $\mu$SR technique. Two sets of polycrystalline LaPt3P specimens, referred to here as sample-A (from Warwick, UK) and sample-B (from ETH, Switzerland), were synthesized at two different laboratories by completely different methods (see Supplementary Note 1 and 2).

Zero-field (ZF), longitudinal-field (LF) and transverse-field (TF) $\mu$SR measurements were performed on these samples at two different muon facilities: sample-A in the MUSR spectrometer at the ISIS Pulsed Neutron and Muon Source, UK, and sample-B in the LTF spectrometer at the Paul Scherrer Institut (PSI), Switzerland.

**ZF-$\mu$SR results.** ZF-$\mu$SR measurements reveal spontaneous magnetic fields arising just below $T_c \approx 1.1$ K (example characterization is shown by the zero-field-cooled magnetic susceptibility ($\chi$) data for sample-B on the right axis of Fig. 1b) associated with a TRS-breaking superconducting state in both samples of LaPt3P, performed on different instruments. Figure 1a shows representative ZF-$\mu$SR time spectra of LaPt3P collected at 75 mK (superconducting state) and at 1.5 K (normal state) on sample-A at ISIS. The data below $T_c$ show a clear increase in muon-spin relaxation rate compared to the data collected in the normal state. To unravel the origin of the spontaneous magnetism at low temperature, we collected ZF-$\mu$SR time spectra over a range of temperatures across $T_c$ and extracted temperature dependence of the muon-spin relaxation rate by fitting the data with a Gaussian Kubo-Toyabe relaxation function $\mathcal{G}(t)$[17] multiplied by an exponential decay:

$$A(t) = A(0)\mathcal{G}(t)\exp(-\lambda_{ZF}t) + A_{bg} \qquad (1)$$

where, $A(0)$ and $A_{bg}$ are the initial and background asymmetries of the ZF-$\mu$SR time spectra, respectively. $\mathcal{G}(t) = \frac{1}{3} + \frac{2}{3}\left(1 - \sigma_{ZF}^2 t^2\right)\exp\left(-\sigma_{ZF}^2 t^2/2\right)$. $\sigma_{ZF}$ and $\lambda_{ZF}$ represent the muon-spin relaxation rates originating from the presence of nuclear and electronic moments in the sample, respectively. The signal-to-background ratio $A(0)/A_{bg} \approx 0.40$ ($\approx 0.52$) for sample-A

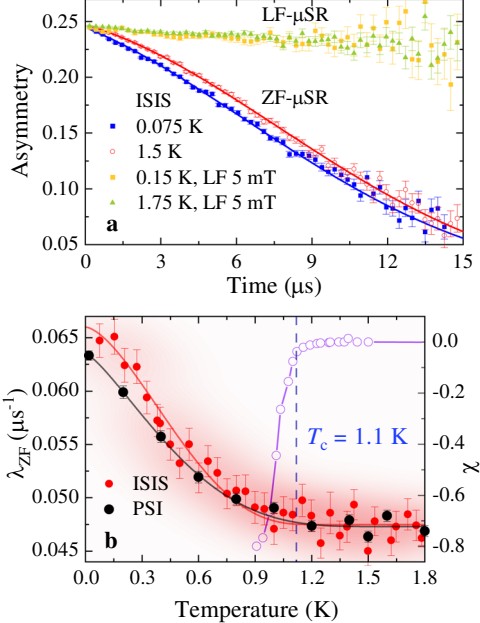

**Fig. 1 Evidence of TRS-breaking superconductivity in LaPt3P by ZF-$\mu$SR measurements. a** ZF-$\mu$SR time spectra collected at 75 mK and 1.5 K for sample-A of LaPt3P. The solid lines are the fits to the data using Eq. (1). **b** The temperature dependence of the extracted $\lambda_{ZF}$ (left axis) for sample-A (ISIS) and sample-B (PSI) showing a clear increase in the muon-spin relaxation rate below $T_c$. The PSI data have been shifted by 0.004 $\mu$s$^{-1}$ to match the baseline value of the ISIS data. Variation of the zero-field-cooled magnetic susceptibility ($\chi$) on the right axis for sample-B. The error bars in **a** and **b** show the standard deviations in the respective measurements.

(sample-B). In the fitting, $\sigma_{ZF}$ is found to be nearly temperature independent and hence fixed to the average value of 0.071(4) $\mu s^{-1}$ for sample-A and 0.050(3) $\mu s^{-1}$ for sample-B. The temperature dependence of $\lambda_{ZF}$ is shown in Fig. 1b. $\lambda_{ZF}$ has a distinct systematic increase below $T_c$ for both the samples which implies that the effect is sample and spectrometer independent. Moreover, the effect can be suppressed very easily by a weak longitudinal-field of 5 mT for both the samples as shown in Fig. 1a for sample-A. This strongly suggests that the additional relaxation below $T_c$ is not due to rapidly fluctuating fields[18], but rather associated with very weak fields which are static or quasistatic on the time-scale of muon life-time. The spontaneous static magnetic field arising just below $T_c$ is so intimately connected with superconductivity that we can safely say its existence is direct evidence for TRS-breaking superconducting state in LaPt$_3$P. From the change $\Delta\lambda_{ZF} = \lambda_{ZF}(T \approx 0) - \lambda_{ZF}(T > T_c)$ we can estimate the corresponding spontaneous internal magnetic field at the muon site $B_{int} \approx \Delta\lambda_{ZF}/\gamma_\mu = 0.22(4)$ G for sample-A and 0.18(2) G for sample-B, which are very similar to that of other TRS-breaking SCs[19]. Here, $\gamma_\mu/(2\pi) = 13.55$ kHz/G is the muon gyromagnetic ratio.

**TF-$\mu$SR results.** We have shown the TF-$\mu$SR time spectra for sample-A in Fig. 2a and Fig. 2b at two different temperatures. The spectrum in Fig. 2a shows only weak relaxation mainly due to the transverse (2/3) component of the weak nuclear moments present in the material in the normal state at 1.3 K. In contrast, the spectrum in Fig. 2b in the superconducting state at 70 mK shows higher relaxation due to the additional inhomogeneous field distribution of the vortex lattice, formed in the superconducting mixed state of LaPt$_3$P. The spectra are analyzed using the Gaussian damped spin precession function[17]:

$$A_{TF}(t) = A(0) \exp\left(-\sigma^2 t^2/2\right) \cos\left(\gamma_\mu \langle B \rangle t + \phi\right) + A_{bg} \cos\left(\gamma_\mu B_{bg} t + \phi\right). \quad (2)$$

Here $A(0)$ and $A_{bg}$ are the initial asymmetries of the muons hitting and missing the sample, respectively. $\langle B \rangle$ and $B_{bg}$ are the internal and background magnetic fields, respectively. $\phi$ is the

initial phase and $\sigma$ is the Gaussian muon-spin relaxation rate of the muon precession signal. The background signal is due to the muons implanted on the outer silver mask where the relaxation rate of the muon precession signal is negligible due to very weak nuclear moments in silver. Figure 2c shows the temperature dependence of $\sigma$ and internal field of sample-A. $\sigma(T)$ shows a change in slope at $T = T_c$, which keeps on increasing with further lowering of temperature. Such an increase in $\sigma(T)$ just below $T_c$ indicates that the sample is in the superconducting mixed state and the formation of vortex lattice has created an inhomogeneous field distribution at the muon sites. The internal fields felt by the muons show a diamagnetic shift in the superconducting state of LaPt$_3$P, a clear signature of bulk superconductivity in this material. The decrease in the internal fields with decreasing temperature below $T_c$ is an indication of a singlet super-conducting ground state.

The true contribution of the vortex lattice field distribution to the relaxation rate $\sigma_{sc}$ can be estimated as $\sigma_{sc} = (\sigma^2 - \sigma_{nm}^2)^{1/2}$, where $\sigma_{nm} = 0.1459(4) \mu s^{-1}$ is the nuclear magnetic dipolar contribution assumed to be temperature independent and was determined from the high-temperature fits. Within the Ginzburg-Landau theory of the vortex state, $\sigma_{sc}$ is related to the London penetration depth $\lambda$ of a SC with high upper critical field by the Brandt equation[20]:

$$\frac{\sigma_{sc}(T)}{\gamma_\mu} = 0.06091 \frac{\Phi_0}{\lambda^2(T)}, \quad (3)$$

where $\Phi_0 = 2.068 \times 10^{-15}$ Wb is the flux quantum. The super-fluid density $\rho \propto \lambda^{-2}$. Figure 3 shows the temperature dependence of $\rho$ normalized by its zero-temperature value $\rho_0$ for LaPt$_3$P. It clearly varies with temperature down to the lowest temperature 70 mK and shows a linear increase below $T_c/3$. This non-constant low temperature behaviour is a signature of nodes in the superconducting gap.

The pairing symmetry of LaPt$_3$P can be understood by analysing the superfluid density data using different models of the gap function $\Delta_\mathbf{k}(T)$. For a given pairing model, we compute the superfluid density ($\rho$) as

$$\rho = 1 + 2\left\langle \int_{\Delta_\mathbf{k}(T)}^{\infty} \frac{E}{\sqrt{E^2 - |\Delta_\mathbf{k}(T)|^2}} \frac{\partial f}{\partial E} dE \right\rangle_{FS}. \quad (4)$$

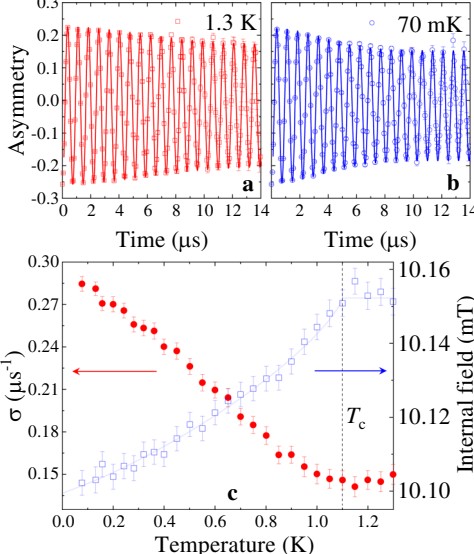

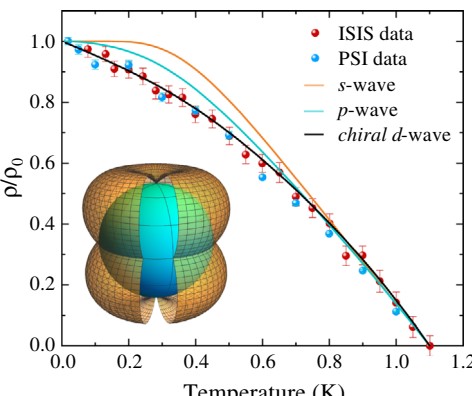

**Fig. 2 Superconducting properties of LaPt$_3$P by TF-$\mu$SR measurements.** TF-$\mu$SR time spectra of LaPt$_3$P collected at **a** 1.3 K and **b** 70 mK for sample-A in a transverse field of 10 mT. The solid lines are the fits to the data using Eq. (2). **c** The temperature dependence of the extracted $\sigma$ (left panel) and internal field (right panel) of sample-A. The error bars show the standard deviations in the TF-$\mu$SR measurements.

**Fig. 3 Evidence of chiral d-wave superconductivity in LaPt$_3$P.** Superfluid density ($\rho$) of LaPt$_3$P as a function of temperature normalized by its zero-temperature value $\rho_0$. The solid lines are fits to the data using different models of gap symmetry. Inset shows the schematic representation of the nodes of the chiral d-wave state. The error bars show the standard deviations in the TF-$\mu$SR measurements in the respective instruments.

Here, $f = 1/\left(1 + e^{\frac{E}{k_B T}}\right)$ is the Fermi function and $\langle\rangle_{FS}$ represents an average over the Fermi surface (assumed to be spherical). We take $\Delta_{\mathbf{k}}(T) = \Delta_m(T)g(\mathbf{k})$ where we assume a universal temperature dependence $\Delta_m(T) = \Delta_m(0) \tanh\left[1.82\left\{1.018\left(T_c/T - 1\right)\right\}^{0.51}\right]$ [21] and the function $g(\mathbf{k})$ contains its angular dependence. We use three different pairing models: s-wave (single uniform superconducting gap), p-wave (two point nodes at the two poles) and chiral d-wave (two point nodes at the two poles and a line node at the equator as shown in the inset of Fig. 3). The fitting parameters are given in the Supplementary Table 2. We note from Fig. 3 that both the s-wave and the p-wave models lead to saturation in $\rho$ at low temperatures, which is clearly not the case for LaPt$_3$P and the chiral d-wave model gives an excellent fit down to the lowest temperature. Nodal SCs are rare since the SC can gain condensation energy by eliminating nodes in the gap. Thus the simultaneous observation of nodal and TRS-breaking superconductivity makes LaPt$_3$P a unique material.

## Discussion

We investigate the normal state properties of LaPt$_3$P by a detailed band structure calculation using density functional theory within the generalized gradient approximation consistent with previous studies[15,22]. LaPt$_3$P is centrosymmetric with a paramagnetic normal state respecting TRS. It has significant effects of spin-orbit coupling (SOC) induced band splitting near the Fermi level (~120 meV, most apparent along the MX high symmetry direction, see Supplementary Note 4). Kramer's degeneracy survives in the presence of strong SOC due to centrosymmetry and SOC only produces small deformations in the Fermi surfaces[23]. The shapes of the Fermi surfaces play an important role in determining the thermodynamic properties of the material. The projections of the four Fermi surfaces of LaPt$_3$P on the $y-z$ and $x-y$ plane are shown in Fig. 4a and Fig. 4b, respectively, with the Fermi surface sheets having the most projected-DOS at the Fermi level shown in blue and orange. It shows the multi-band nature of LaPt$_3$P with orbital contributions mostly coming from the $5d$ orbitals of Pt and the $3p$ orbitals of P.

LaPt$_3$P has a non-symmorphic space group P4/mmm (No. 129) with point group D$_{4h}$. From the group theoretical classification of the SC order parameters within the Ginzburg-Landau theory[19,24], the only possible superconducting instabilities with strong SOC, which can break TRS spontaneously at $T_c$ correspond to the two 2D irreducible representations, $E_g$ and $E_u$, of D$_{4h}$. Non-symmorphic symmetries can give rise to additional symmetry-required nodes on

the Brillouin zone boundaries along the high symmetry directions. The non-symmorphic symmetries of LaPt$_3$P, however, can only generate additional point nodes for the $E_g$ order parameter but no additional nodes for the $E_u$ case[25]. The superconducting ground state in the $E_g$ channel is a pseudospin chiral d-wave singlet state with gap function $\Delta(\mathbf{k}) = \Delta_0 k_z(k_x + ik_y)$ where $\Delta_0$ is a complex amplitude independent of $\mathbf{k}$. The $E_u$ order parameter is a pseudospin nonunitary chiral p-wave triplet state with d-vector $\mathbf{d}(\mathbf{k}) = [c_1 k_z, ic_1 k_z, c_2(k_x + ik_y)]$ where $c_1$ and $c_2$ are material dependent real constants independent of $\mathbf{k}$.

We compute the quasi-particle excitation spectrum for the two TRS-breaking states on a generic single-band spherical Fermi surface using the Bogoliubov-de Gennes mean-field theory[19,24]. The chiral d-wave singlet state leads to an energy gap given by $|\Delta_0||k_z|(k_x^2 + k_y^2)^{1/2}$. It has a line node at the "equator" for $k_z = 0$ and two point nodes at the "north" and "south" poles (shown in Fig. 4a). The low temperature thermodynamic properties are, however, dominated by the line node because of its larger low energy DOS than the point nodes. The triplet state has an energy gap given by $[g(k_x, k_y) + 2c_1^2 k_z^2 - 2|c_1||k_z|\{f(k_x, k_y) + c_1^2 k_z^2\}^{1/2}]^{1/2}$ where $f(k_x, k_y) = c_2^2(k_x^2 + k_y^2)$. It has only two point nodes at the two poles and no line nodes. Thus, the low temperature linear behaviour of the superfluid density of LaPt$_3$P shown in Fig. 3 is only possible in the chiral d-wave state with a line node in contrast to the triplet state with only point nodes, which will give a quadratic behaviour and saturation at low temperatures.

The preceding discussion assuming a generic Fermi surface can be adapted for the case of the inherently multi-band material LaPt$_3$P by considering the momentum dependence of the gap on the Fermi surfaces sheets neglecting interband pairing. We note from Fig. 4a and Fig. 4b that there are two important Fermi surface sheets in LaPt$_3$P, with the chiral d-wave state having the two point nodes on one of the Fermi surface sheets and a line node on the other. Thus LaPt$_3$P is one of the rare unconventional SCs for which we can unambiguously identify the superconducting order parameter.

The severe constraints on the possible pairing states as a result of the unique properties of LaPt$_3$P lead us to expect that our experimental observations will be consistent only with a chiral d-wave like order parameter belonging to the $E_g$ channel even after considering pairing between bands in a multi-orbital picture[10]. It is also intriguing to think about the possible pairing mechanism giving rise to the chiral d-wave state in this material, which has a weakly correlated normal state, weak electron–phonon coupling and no spin fluctuations[15,16]. These issues will be taken up in future investigations.

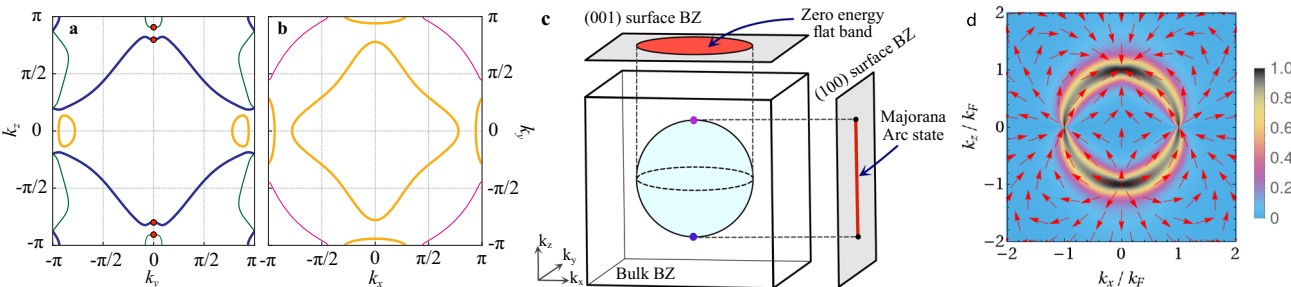

**Fig. 4 Properties of the normal and superconducting states of LaPt$_3$P.** Projections of the four Fermi surfaces of LaPt$_3$P with SOC on the $y-z$ plane in **a** and $x-y$ plane in **b**. The thickness of the lines are proportional to the contribution of the Fermi surfaces to the DOS at the Fermi level (green—10.3%, blue—43.4%, orange—40% and magenta—6.3%). The point nodes of the chiral d-wave gap are shown by red dots in **a** and the line node resides on the $x-y$ plane in **b**. **c** Schematic view of the Majorana Fermi arc and the zero-energy Majorana flat band corresponding to the two Weyl point nodes and the line node respectively on the respective surface Brillouin zones (BZs) assuming a spherical Fermi surface. **d** Berry curvature $\mathbf{F}(\mathbf{k})$ corresponding to the two Weyl nodes on the $x-z$ plane. Arrows show the direction of $\mathbf{F}(\mathbf{k})$ and the colour scale shows its magnitude $= \frac{2}{\pi}\arctan(|\mathbf{F}(\mathbf{k})|)$. $\Delta_0 = 0.5\,\mu$ was chosen for clarity while a more realistic weak-coupling limit $\Delta_0 \ll \mu$ gives a more sharply peaked curvature at the Fermi surface.

The topological properties of the chiral $d$-wave state of $LaPt_3P$ are most naturally discussed considering a generic single-band spherical Fermi surface (chemical potential $\mu = k_F^2/(2m)$ where $k_F$ is the Fermi wave vector and $m$ is the electron mass)[4,26]. However, topological protection of the nodes[27] also ensures stability against multi-band effects assuming interband pairing strengths to be small. The effective angular momentum of the Cooper pairs is $L_z = +1$ (in units of $\hbar$) with respect to the chiral $c$-axis. The equatorial line node acts as a vortex loop in momentum space[28] and is topologically protected by a 1D winding number $w(k_x, k_y) = 1$ for $k_x^2 + k_y^2 < k_F^2$ and $= 0$ otherwise. The non-trivial topology of the line node leads to two-fold degenerate zero-energy Majorana bound states in a flat band on the $(0, 0, 1)$ surface BZ as shown in Fig. 4c. As a result, there is a diverging zero-energy DOS leading to a zero-bias conductance peak (which can be really sharp[29]) measurable in STM. This inversion symmetry protected line node is extra stable due to even parity SC[29,30]. The point nodes on the other hand are Weyl nodes and are impossible to gap out by symmetry-preserving perturbations. They act as a monopole and an anti-monopole of Berry flux as shown in Fig. 4d and are characterized by a $k_z$-dependent topological invariant, the sliced Chern number $C(k_z) = L_z$ for $|k_z| < k_F$ with $k_z \neq 0$ and $= 0$ otherwise (see Supplementary Note 6 for details). As a result, the $(1, 0, 0)$ and $(0, 1, 0)$ surface BZs each have a Majorana Fermi arc, which can be probed by STM as shown in Fig. 4c. There are two-fold degenerate chiral surface states with linear dispersion carrying surface currents leading to local magnetisation that can be detected using SQUID magnetometry. One of the key signatures of chiral edge states is the anomalous thermal Hall effect (ATHE), which depends on the length of the Fermi arc in this case. Impurities in the bulk can, however, increase the ATHE signal by orders of magnitude[31] over the edge contribution making it possible to detect with current experimental technology[32]. We also note that a 90° rotation around the $c$-axis for the chiral $d$-wave state leads to a phase shift of $\pi/2$, which can be measured by corner Josephson junctions[33].

## Methods

**$\mu$SR technique**. $\mu$SR is a very sensitive microscopic probe to detect the local-field distribution within a material. This technique has been widely used to search for very weak fields (of the order of a fraction of a gauss) arising spontaneously in the superconducting state of TRS-breaking SCs. The other great use of this technique is to measure the value and temperature dependence of the London magnetic penetration depth, $\lambda$, in the vortex state of type-II SCs[34]. $1/\lambda^2(T)$ is in turn proportional to the superfluid density, which can provide direct information on the nature of the superconducting gap. Details of the $\mu$SR technique is given in Supplementary Note 3.

**Sample preparation and characterisation**. Two sets of polycrystalline samples (referred to as sample-A and sample-B) of $LaPt_3P$ were synthesized at two different laboratories (Warwick, UK and PSI, Switzerland) by completely different methods. While, sample-A was synthesized by solid state reaction method, sample-B was synthesized using the cubic anvil high-pressure and high-temperature technique. Details of the sample preparation and characterization are given in Supplementary Note 1 and 2.

**DFT calculation**. The first principles density functional theory (DFT) calculations were performed by the full potential linearized augmented plane wave method implemented in the WIEN2k package[35]. The generalized gradient approximation with the Perdew-Burke-Ernzerhof realization was used for the exchange-correlation functional. The plane wave cut-off $K_{max}$ is given by $R_{mt} * K_{max} = 8.0$. For the self-consistent calculations, the BZ integration was performed on a $\Gamma$-centred mesh of $15 \times 15 \times 15$ k-points.

## Data availability

All the datasets that support the findings of this study are available from the corresponding author upon reasonable request. The ISIS DOI for our MUSR source data is https://doi.org/10.5286/ISIS.E.RB1720467.

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

## Acknowledgements

P.K.B. gratefully acknowledges the ISIS Pulsed Neutron and Muon Source of the UK Science & Technology Facilities Council (STFC) and Paul Scherrer Institut (PSI) in Switzerland for access to the muon beamtimes. S.K.G. thanks Jorge Quintanilla and Adhip Agarwala for stimulating discussions and acknowledges the Leverhulme Trust for support through the Leverhulme early career fellowship. The work at the University of Warwick was funded by EPSRC,UK, Grant EP/T005963/1. X.X. was supported by the National Natural Science Foundation of China (Grant 11974061). N.D.Z. thanks K. Povarov and acknowledges support from the Laboratory for Solid State Physics, ETH Zurich where synthesis studies were initiated.

## Author contributions

P.K.B. conceived the project, successfully acquired the PSI and ISIS muon beamtimes and performed the µSR experiments and data analysis. S.K.G. performed the theory part of the project using band structure calculations performed by J.Z.Z. S.K.G. helped in data analysis and wrote the manuscript together with P.K.B. D.A.M. synthesized and characterized the sample from Warwick and participated in the µSR experiments at ISIS. N.D.Z. synthesized and characterized the sample from ETH. C.B. helped in performing the µSR experiments in PSI. X.X., A.D.H., G.B. and M.R.L. helped to improve the presentation of the data and the manuscript as a whole.

## Competing interests

The authors declare no competing interests.
