## [Peer Review File · Nature Communications]

REVIEWER COMMENTS

Reviewer #1 (Remarks to the Author):

Biswas and collaborators report μ SR measurements and extract the superfluid-density of the superconductor LaPt₃P which was discovered several years ago. The measurements lead to the conclusion that a chiral d-wave singlet superconducting state is realized in LaPt₃P.

To start with, it has to be said that this is an exciting discovery, were it to be found true. I do not know of any material that has unambiguously been verified to have such order parameter. This then also puts the task of providing full evidence on the authors. I am a bit hesitating to affirm that the authors have provided full evidence (see further below).

To start with, the abstract refers to UPt₃ and Sr₂RuO₄ as f- and p- wave SCs. Recent measurements have shown that Sr₂RuO₄ is no longer considered as a p-wave triplet SC. UPt₃ is still considered unconventional; here it is needed to cite a proper review on this material, that provides and discusses the experimental evidence. The connection to the first line of the abstract is weak (topological superconductors with Majorana bound states); it could be removed as the abstract is quite long and the focus is on the outcome of the bulk measurements.

The introduction again mentions Sr₂RuO₄, but the unconventional character is no longer substantiated.

The subsequent paragraph introduces the platinum pnictide family. These have been discussed in the literature (theory and experiment) in the last years. The authors write that Migdal-Eliashberg theory gave indications of the unconventional nature of SC in LaPt₃P. This seems to be rather an interpretation of the data given in ref. 12, see also the recent paper by Aperis et al, Ann. Phys. 417, 168095 (2020). The conclusion in these two works is that LaPt₃P is the most BCS-like material in this family with a weak electron-phonon coupling constant λ . The linear-temperature specific heat (C/T vs. T) is maybe not well described, but the measurements did not extend sufficiently below 1 K to be conclusive. It is thus difficult to see an indication of unconventional SC in LaPt₃P from these works.

This should be presented better in this paragraph. Also, to address the low-temperature specific heat in the SC state, can the authors offer better measurements of this quantity to support further their claims (e.g. using the data shown in the SM)?

The μ SR measurements are in themselves presented clearly. Can the authors mention the position of the muon the unit cell, how is it known, and if there is possibly an anisotropy of the internal field?

Also, some of the authors previously used μ SR to show superconductivity with time-reversal symmetry breaking in SrPtAs, suggesting also a chiral d-wave order parameter for this related material. However, one of the explanations was a granularity scenario. Can this be excluded completely here for LaPt₃P? This should be discussed better and in a conclusive way.

Although the claim of chiral d-wave superconductivity is exciting, the manuscript does not offer any reasoning or explanation of the pairing mechanism. The electron-phonon coupling is weak (but explained the small T_c) and electron-electron correlations are also weak. Spin fluctuations are not present in the material. What could then be the unconventional pairing mechanism?

The data shown in Fig. 3 support chiral d-wave superconductivity. The authors honestly write that the results are for a spherical Fermi surface. The real Fermi surface has several anisotropic FS sheets. The question is thus what would happen when one would use a realistic anisotropic two- or three-band

model. The claim of chiral d-wave symmetry is important but there are chances that the data can be explained by something else.

The topological analysis at the end of the manuscript is nice but a bit general (note that the spherical Fermi surface is assumed). The connection to the specific material is not strong that would render this part particularly interesting.

Lastly, some of the plots in the SM should be improved. I could hardly see what is in the plots (Figs. 5 and 7).

Summarizing, an unambiguous discovery of chiral d-wave singlet SC in LaPt₃P would be exciting, but this has not yet been reached in the current version of the manuscript. It is requested that the authors substantiate their claims further, in order to provide convincing evidence.

Reviewer #2 (Remarks to the Author):

I enjoyed reading this manuscript. The authors present μ SR data and theoretical calculations that demonstrate the existence of a chiral d-wave singlet superconducting state in LaPt₃P. Experimentally, the μ SR data show that time reversal symmetry is broken in the superconducting state and that the superfluid density decreases linearly with temperature increasing from $T = 0$, indicating that the superconducting order parameter possesses line nodes. Theoretically, the authors show that based on the known symmetry of this system, the only superconducting state consistent with these experimental observations is a chiral d-wave singlet state, making this is a rare example of a topological superconductor with robustly protected Majorana modes that could potentially be useful for quantum computation. The paper is straightforward and convincing (in my view), and I recommend publication in Nature Communications, provided the authors can adequately address the points below.

1. In the second paragraph on the first page, the authors state without reference that unconventional superconductors are defined as having "zero average onsite pairing amplitude". I do not believe that this will be understandable for the general audience of Nature Communications, nor do I believe that this is a commonly used definition of unconventional superconductors. The authors should provide additional clarification and references.
2. In the first paragraph of the experimental results, I would recommend including a reference to the supplementary information immediately after the statement that the two samples were synthesized by two different methods so that the readers know where to go for more information.
3. The word "characterization" is misspelled as "charcterization" on p. 1.
4. I would request the authors to include equivalent versions of Fig. 1a, 2a, 2b, and 2c using Sample B in the supplementary information.
5. The authors should explicitly state which parameters in Eq. 1 and Eq. 2 were temperature independent and which were temperature dependent in the fitting. Additionally, the values of $A(0)$ and Abg should be stated so readers know what the signal-to-background ratio is.
6. Why is σ_{ZF} different for the two samples? Assuming that this term arises only from the nuclear dipolar moments and that the muons have the same stopping site(s) in each sample, one would expect both samples to have the same value.

7. Near the top of p. 3, the authors should clarify that the value of σ_{nm} was determined from the high-temperature fits, which show a plateau at this value.
8. On p. 3, the authors state that Fig. 3 shows the superfluid density ρ , when in reality it shows the normalized quantity $\rho/\rho(T=0)$. For accuracy, the authors should correct this in the text.
9. The theoretical calculation of ρ , and much of the following theoretical discussion, assumes that the Fermi surface is spherical. However, the calculated Fermi surface is not spherical, as shown in Fig. 4 and the supplementary information. The authors therefore need to justify why the assumption of a spherical Fermi surface is valid. Would it be possible to implement Eq. 4 using the theoretically calculated Fermi surface, or is that not feasible?
10. The sentence on p. 4 beginning with "While the Eu order parameter..." is a sentence fragment. The authors should remove the word "While".
11. Stylistically, I do not like the line in the second column of p. 4 that begins with "gap = $\sqrt{(\dots)}$ ". In place of the equals sign, perhaps the authors could write "equal to" or "given by" or "of."

“Chiral singlet superconductivity in the weakly correlated metal LaPt₃P”

A. Summary of Changes

We thank the referees for carefully considering our manuscript. We have taken into account all of their suggestions in the revised version of the manuscript. The changes are clearly marked in blue in the attached PDFs of the main text and the supplementary.

B. Response to the referee reports

Reviewer #1 (Remarks to the Author)

Biswas and collaborators report μ SR measurements and extract the superfluid-density of the superconductor LaPt₃P which was discovered several years ago. The measurements lead to the conclusion that a chiral d-wave singlet superconducting state is realized in LaPt₃P.

To start with, it has to be said that this is an exciting discovery, were it to be found true. I do not know of any material that has unambiguously been verified to have such order parameter. This then also puts the task of providing full evidence on the authors. I am a bit hesitating to affirm that the authors have provided full evidence (see further below).

To start with, the abstract refers to UPt₃ and Sr₂RuO₄ as f- and p- wave SCs. Recent measurements have shown that Sr₂RuO₄ is no longer considered as a p-wave triplet SC. UPt₃ is still considered unconventional; here it is needed to cite a proper review on this material, that provides and discusses the experimental evidence.

Response: We thank the referee for pointing this out. We no longer refer to UPt₃ and Sr₂RuO₄ in the abstract. Instead we discuss these materials in the introduction by citing additional relevant references clarifying the current understanding of the physics of these materials. (page-1, 2nd para, refs. 9-13 are newly added)

The connection to the first line of the abstract is weak (topological superconductors with Majorana bound states); it could be removed as the abstract is quite long and the focus is on the outcome of the bulk measurements.

Response: We have now amended this according to the suggestion of the referee. (see the abstract)

The introduction again mentions Sr₂RuO₄, but the unconventional character is no longer substantiated.

Response: We discuss the current understanding of the nature of superconductivity in Sr₂RuO₄ and cite the relevant references in the introduction. (page-1, 2nd para)

The subsequent paragraph introduces the platinum pnictide family. These have been discussed in the literature (theory and experiment) in the last years. The authors write that Migdal-Eliashberg theory gave indications of the unconventional nature of SC in LaPt₃P. This seems to be rather an interpretation of the data given in ref. 12, see also the recent paper by

Aperis et al, Ann. Phys. 417, 168095 (2020). The conclusion in these two works is that LaPt₃P is the most BCS-like material in this family with a weak electron-phonon coupling constant λ . The linear-temperature specific heat (C/T vs. T) is maybe not well described, but the measurements did not extend sufficiently below 1 K to be conclusive. It is thus difficult to see an indication of unconventional SC in LaPt₃P from these works. This should be presented better in this paragraph.

Response: We thank the referee for highlighting these results. We have now added a few sentences in this paragraph to present these results in an improved and more clear manner. (page-2, first para, first column)

Also, to address the low-temperature specific heat in the SC state, can the authors offer better measurements of this quantity to support further their claims (e.g. using the data shown in the SM)?

Response: We have measured specific heat on several batches of the polycrystalline samples of LaPt₃P as shown below but did not get any better-quality data (e.g. using the data shown in the SM). Similar to the other report in Takayama et al., PRL 108, 237001 (2012), the heat capacity anomaly at T_c in LaPt₃P is very weak, which is possibly masked by a hyperfine contribution at low temperatures. However, the bulk superconductivity is confirmed by both the large shielding volume fraction in the magnetization measurement (see Fig. 1b of the main text) and the diamagnetic shift seen in the internal field measured in transverse-field μ SR experiments (see Fig. 2C of the main text). We are now trying to grow single crystals for this material with much effort and if successful, it will certainly offer us a better chance to obtain not only high-quality low-temperature specific heat data, but also data for other relevant quantities. To grow the single crystals of this type of compounds proves to be extremely difficult and as such, there is no report on single crystals of LaPt₃P in the literature so far. It is, however, out of scope of the current project mainly due to the uncertainty and time required in growing good quality single crystals, and perform the measurements thereafter.

Fig. R1: The heat capacity data measured on two batches of polycrystalline sample-A of LaPt₃P.

The muSR measurements are in themselves presented clearly. Can the authors mention the position of the muon the unit cell, how is it known, and if there is possibly an anisotropy of the internal field?

Response: When a sufficiently large field (higher than the lower critical field, H_{c1}) is applied to a type-II superconductor, the field enters inside the superconductor in the form of lines of flux, which are spread out over an area of dimension λ , the magnetic penetration depth of the superconductor. The magnetic field will be non-uniform, being highest at the flux line cores. Then, the implanted muons, although at a definite position within the unit cell, may be considered to arrive at random positions within the flux-lattice, since both the spacing of flux lines (~ 100 nm at a typical $B \sim 0.2$ T) and the value of λ (~ 650 nm) are much larger than the unit cell size. Hence the muons should obtain an unbiased sample position of the mixed-state field distribution. Muons implanted at random positions with respect to the superconducting flux line lattice, remain static after implantation. They do not affect the properties of the surrounding superconductor appreciably and will experience a range of different fields and precess at different rates, leading to a damping of the muon-spin-rotation signal.

For highly anisotropic single crystal samples such as Fe-based and Cuprate high- T_c layered superconductors, one would expect to see some level of anisotropy in the internal field. Given that the crystal structure of LaPt_3P is nearly 3-dimensional and we measured polycrystalline samples, it is highly unlikely that the internal field will be anisotropic. Please see below Fourier transform of muSR time spectra showing isotropic Gaussian field distribution of the internal fields in the normal state as well as in the superconducting state.

Fig. R2: Fourier transform of muSR time spectra showing isotropic Gaussian field distribution of the internal fields in the normal state (2 K) as well as in the superconducting state (19 mK).

Also, some of the authors previously used muSR to show superconductivity with time-reversal symmetry breaking in SrPtAs, suggesting also a chiral d-wave order parameter for this related material. However, one of the explanations was a granularity scenario. Can this be excluded completely here for LaPt_3P ? This should be discussed better and in a conclusive way.

Response: We thank the referee for bringing up this important issue. However we can exclude this possibility for the case of LaPt₃P as argued below.

Biswas et al. in PRB 87, 180503(R) (2013) reported that for SrPtAs polycrystalline samples the internal field measured by TF- μ SR experiment increases with decreasing temperature below T_c . One possible explanation for this unusual positive field shift is the so called granularity scenario. In this proposed mechanism due to granularity of the samples two or more crystallites can form frustrated current loops giving positive field shifts at low fields. In this mechanism the bulk remains time-reversal symmetry (TRS) invariant and the boundary breaks TRS. The relevant instability is a f-wave triplet belonging to the A_{2u} channel. However, TRS breaking requires it to be degenerate with the instability in the A_{2g} channel. As a result there are two transitions in this scenario and TRS breaking happens only below the lower transition which is *different* from T_c .

Now, for the case of LaPt₃P, Fig.2b of main text (sample A) and Supplementary Fig.6d (sample B) show the variations of the internal field as a function of temperature as measured in TF- μ SR experiments. We note from these figures that the internal fields show a diamagnetic shift below T_c which is a clear indication of bulk superconductivity in this material. Also, the internal fields decrease with decreasing temperature below T_c which is usually an indication of singlet pairing in the system. Moreover, as shown in Fig.1b of the main text, the ZF- μ SR results clearly show TRS breaking at the superconducting T_c and *not* at a separate transition away from T_c . Therefore the granularity scenario is clearly not consistent with our experimental observations for LaPt₃P and can be excluded.

Although the claim of chiral d-wave superconductivity is exciting, the manuscript does not offer any reasoning or explanation of the pairing mechanism. The electron-phonon coupling is weak (but explained the small T_c) and electron-electron correlations are also weak. Spin fluctuations are not present in the material. What could then be the unconventional pairing mechanism?

Response: We thank the referee for raising this important question. As the referee pointed out, it is really intriguing how a chiral d-wave state emerges out of a weakly correlated normal state, with weak electron-phonon coupling and without spin fluctuations. One speculation is that considering the multi-band nature of LaPt₃P might be important. It was shown recently by Suh et al. in Phys. Rev. Research 2, 032023(R) (2020) for the case of Sr₂RuO₄ that a chiral *d*-wave like superconducting ground state can be stabilized as a result of an interplay between an onsite Hubbard interaction, Hund's exchange and pair-hopping interactions in the weak-coupling regime within a multi-orbital picture. A similar scenario can be at play here as well. However, we do not have a reasonable answer to this question at the moment and hopefully our work will motivate more investigations into this interesting question in the community, both theoretically and experimentally.

The data shown in Fig. 3 support chiral d-wave superconductivity. The authors honestly write that the results are for a spherical Fermi surface. The real Fermi surface has several anisotropic FS sheets. The question is thus what would happen when one would use a realistic anisotropic two- or three-band model. The claim of chiral d-wave symmetry is important but there are chances that the data can be explained by something else.

Response: Group theoretical symmetry analysis within the Ginzburg-Landau framework is a powerful tool to find all the possible superconducting

instabilities based only on the symmetries of the normal state of a specific material. However, most of the times this general analysis, even within the single band picture, provides many possible superconducting instability channels with similar low-temperature thermodynamic properties. Thus it is in general difficult to pin-point a particular order parameter for a specific material. LaPt₃P is, however, unique as a result of a combination of its following properties: a) Strong spin-orbit coupling effects as seen from the band structure calculations, b) Broken time-reversal symmetry in the superconducting state at T_c as seen in the zero-field μ SR experiments, c) Bulk superconductivity as can be inferred from the diamagnetic shift seen in the internal field measured in transverse-field μ SR experiments and d) Line nodal behavior of the low-temperature superfluid density measured by transverse-field μ SR experiments. All of these experimental observations can be consistently explained by bulk superconductivity with chiral d-wave pairing symmetry. The multi-band nature of LaPt₃P is accommodated in this picture by simply considering the momentum dependence of the gap corresponding to the chiral d-wave state on the most important Fermi surface sheets of LaPt₃P which contribute the most to the density of states at the Fermi level. As shown in Fig.4a and Fig.4b of the main text, there are two most important Fermi surface sheets. The two point nodes reside on one of them and the line node resides on the other. As a result, the low-temperature thermodynamic properties of LaPt₃P will be dominated by the line node.

To fit the data in Fig.3 of the main text we first use the most symmetric Fermi surface (the spherical Fermi surface) to establish the pairing symmetry of the superconducting state. Then in the later discussion we take the specific case of the LaPt₃P material.

However, as the referee have pointed out we have not considered interband pairing in the discussion and we should consider a realistic multi-band description of the normal state. In this regard, it is to be noted that the unique properties of LaPt₃P as detailed above put severe constraints on the possible pairing state. And, as shown recently for the case of Sr₂RuO₄ in Suh et al. Phys. Rev. Research 2, 032023 (2020), even after considering a multi-orbital system with all the inter-band pairings we expect the order parameter to be of chiral d-wave symmetry belonging to the E_g channel to consistently explain all the experimental observations.

We thank the referee for raising this important point. We have now clarified it in the main text by adding a few sentences and references. (page-5, 1st column, 3rd para)

The topological analysis at the end of the manuscript is nice but a bit general (note that the spherical Fermi surface is assumed). The connection to the specific material is not strong that would render this part particularly interesting.

Response: The topological properties of the chiral d-wave state is most naturally discussed assuming a system with the most symmetric Fermi surface, i.e. a spherical Fermi surface. The connection to the specific material then comes from looking the momentum dependence of the pairing potential on the most important Fermi surfaces. In this case, the two most important Fermi surfaces of LaPt₃P (“blue” and “orange” shown in Fig.4a and Fig.4b respectively) accommodate the two point nodes and the line node respectively. As a consequence of topological protection of the nodes, the Fermi arcs and the zero-energy surface states are expected to be stable against multi-band effects as long as the inter-band pairing strengths are small.

We thank the referee for bringing up this important point. We have now clarified this by rephrasing a sentence and adding a sentence fragment. (page-5, 4th para)

Lastly, some of the plots in the SM should be improved. I could hardly see what is in the plots (Figs. 5 and 7).

Response: We have now updated these figures (see Fig.7 in page-6 and Fig. 9 in page-8 of the SM) according to the referee's suggestion.

Summarizing, an unambiguous discovery of chiral d-wave singlet SC in LaPt₃P would be exciting, but this has not yet been reached in the current version of the manuscript. It is requested that the authors substantiate their claims further, in order to provide convincing evidence.

Response: We thank the referee for a careful consideration of our manuscript. We believe, in the revised version of the manuscript, we have now conclusively demonstrated that all our experimental observations in LaPt₃P can be consistently explained by a chiral d-wave singlet superconducting ground state.

Report of Referee B -- LM17104/Ghosh

Reviewer #2 (Remarks to the Author):

I enjoyed reading this manuscript. The authors present μ SR data and theoretical calculations that demonstrate the existence of a chiral d-wave singlet superconducting state in LaPt₃P. Experimentally, the μ SR data show that time reversal symmetry is broken in the superconducting state and that the superfluid density decreases linearly with temperature increasing from $T = 0$, indicating that the superconducting order parameter possesses line nodes. Theoretically, the authors show that based on the known symmetry of this system, the only superconducting state consistent with these experimental observations is a chiral d-wave singlet state, making this is a rare example of a topological superconductor with robustly protected Majorana modes that could potentially be useful for quantum computation. The paper is straightforward and convincing (in my view), and I recommend publication in Nature Communications, provided the authors can adequately address the points below.

1. In the second paragraph on the first page, the authors state without reference that unconventional superconductors are defined as having "zero average onsite pairing amplitude". I do not believe that this will be understandable for the general audience of Nature Communications, nor do I believe that this is a commonly used definition of unconventional superconductors. The authors should provide additional clarification and references.

Response: We thank the referee for this suggestion. We have now added two sentences to clarify this point and we have also added a reference. (page-1, first paragraph in the introduction, reference 6 is the newly added one)

2. In the first paragraph of the experimental results, I would recommend including a reference to the supplementary information immediately after the statement that the two samples were synthesized by two different methods so that the readers know where to go for more information.

Response: We agree with the referee and we have now given the reference to Supplementary Section I and Section II in the first paragraph of the experimental results section. (page-2, 1st column, 3rd para)

3. The word "characterization" is misspelled as "charcterization" on p. 1.

Response: We have now corrected this spelling mistake. (page-2, 1st column, 4th para)

4. I would request the authors to include equivalent versions of Fig. 1a, 2a, 2b, and 2c using Sample B in the supplementary information.

Response: We have now added these figures in the supplementary information. Please see Fig.5 and Fig.6 in the supplemental material.

5. The authors should explicitly state which parameters in Eq. 1 and Eq. 2 were temperature independent and which were temperature dependent in the fitting. Additionally, the values of $A(0)$ and A_{bg} should be stated so readers know what the signal-to-background ratio is.

Response: We have now added this information in the text. (page-2, 2nd column)

6. Why is σ_{ZF} different for the two samples? Assuming that this term arises only from the nuclear dipolar moments and that the muons have the same stopping site(s) in each sample, one would expect both samples to have the same value.

Response: We agree with the referee that σ_{ZF} should be the same for the two samples. However, since the two samples were prepared by two completely different methods, show slightly different superconducting volume fractions and hence slightly different signal-to-background ratio in the asymmetry spectra, and measured at two different muon instruments, it is not unusual to see a small difference in σ_{ZF} for the two (σ_{ZF} is $0.07/\mu\text{s}$ for sample-A measured at ISIS and $0.05/\mu\text{s}$ for sample-B measured at PSI).

7. Near the top of p. 3, the authors should clarify that the value of σ_{nm} was determined from the high-temperature fits, which show a plateau at this value.

Response: According to the suggestion of the referee we have added this in the text. (page-3, 2nd column, last para)

8. On p. 3, the authors state that Fig. 3 shows the superfluid density ρ , when in reality it shows the normalized quantity $\rho/\rho(T=0)$. For accuracy, the authors should correct this in the text.

Response: As suggested by the referee we have now amended this in the text. (page-4, first para)

9. The theoretical calculation of ρ , and much of the following theoretical discussion, assumes that the Fermi surface is spherical. However, the calculated Fermi surface is not spherical, as shown in Fig. 4 and the supplementary information. The authors therefore need to justify why the assumption of a spherical Fermi surface is valid. Would it be possible to implement Eq. 4 using the theoretically calculated Fermi surface, or is that not feasible?

Response: LaPt₃P is a an inherently multi-band system with two most important Fermi surface sheets (contributing the most to the density of states at the Fermi level) as shown in Fig.4a and Fig.4b of the main text. The theoretical calculation of ρ for the purpose of fitting the experimental data presented in Fig.3 of the main text has been performed assuming a spherical Fermi surface. This is because for the purpose of deciphering the pairing symmetry of the superconducting state the most important thing is to have the correct symmetry of the pairing potential with all its nodal structure. This is most naturally done by assuming the most symmetric normal state i.e. a spherical Fermi surface. Then the connection to the specific material LaPt₃P is that we adapt this analysis by considering the momentum dependence of the gap on the most important Fermi surfaces. For this case, the two point nodes and the line node corresponding to the chiral d-wave state reside on the two most important Fermi surfaces respectively. As a result the low-temperature thermodynamic properties of LaPt₃P will be dominated by the line node explaining the low temperature linear behavior of the superfluid density. As the referee pointed out it is possible to use the k-points corresponding to the Fermi surface sheets in doing the average in Eq.4 of the main text assuming only intraband pairing and phenomenologically using the pairing potential. However, it will reproduce the linear low-temperature behavior because of the higher low energy density of states coming from the line node residing on one of the important Fermi surface sheets.

The subsequent discussion of topological properties of the pairing state (chiral d-wave in this case) is most naturally illustrated using a spherical Fermi surface. The connection to the specific material then follows similarly by considering the momentum dependence of the pairing potential on the most important Fermi surface sheets. The topological protection of the nodes ensures stability of the surface states against multiband effects as long as the interband pairing strengths are small.

We have now clarified these points in the main text by adding a few sentences and citations. (page-5, 3rd and 4th paras, ref. 13 and 30 are newly added ones)

10. The sentence on p. 4 beginning with "While the Eu order parameter..." is a sentence fragment. The authors should remove the word "While".

Response: We have now amended this according to the referee's suggestion. (page-5, 1st para)

11. Stylistically, I do not like the line in the second column of p. 4 that begins with "gap = $\sqrt{(\dots)}$ ". In place of the equals sign, perhaps the authors could write "equal to" or "given by" or "of."

Response: We have now amended these according to the referee's suggestion. (page-5, 2nd para)

REVIEWERS' COMMENTS

Reviewer #1 (Remarks to the Author):

The authors have satisfactorily addressed the comments that I had in my previous report. The microscopic origin of the time-reversal symmetry-breaking superconductivity remains unfortunately unclear. It is therefore desirable that the authors add a few lines on this to the manuscript.

As a further question, there seems to be an anomaly in the specific heat at 1.5 K (above T_c , Fig. 2 in SM) which isn't discussed. The authors should add a sentence on this to the SM.

When those last things have been addressed, I believe that the manuscript is recommendable for publication in Nature Communications.

Reviewer #2 (Remarks to the Author):

In my opinion, the authors have improved the manuscript and satisfactorily addressed the concerns raised during the review process. I recommend publishing this article in Nature Communications.

“Chiral singlet superconductivity in the weakly correlated metal LaPt₃P”

A. Summary of Changes

We thank the referees for carefully considering our resubmitted manuscript and recommending it for publishing in Nature communications. We have taken into account the two suggestions made by Reviewer-1 in the revised version of the manuscript and we note that Reviewer-2 has not suggested any further changes. The changes are clearly marked in blue in the attached PDFs of the main text and the supplementary.

B. Response to the referee reports

Reviewer #1 (Remarks to the Author)

The authors have satisfactorily addressed the comments that I had in my previous report.

The microscopic origin of the time-reversal symmetry-breaking superconductivity remains unfortunately unclear. It is therefore desirable that the authors add a few lines on this to the manuscript.

Response: We have added two lines clearly stating this point in the manuscript. (page-5, 1st column, 3rd paragraph of the manuscript)

As a further question, there seems to be an anomaly in the specific heat at 1.5 K (above T_c, Fig. 2 in SM) which isn't discussed. The authors should add a sentence on this to the SM.

Response: We have now added two sentences discussing this point in the SM. (page-3, 1st paragraph of SM)

When those last things have been addressed, I believe that the manuscript is recommendable for publication in Nature Communications.

Response: We are grateful to the referee for accepting our paper.